# Crescent Microstrip Antenna for LTE-U and 5G Systems

**Rafał Przesmycki \*** and **Marek Bugaj**

Department of Electronics, Military University of Technology, 00-908 Warsaw, Poland; marek.bugaj@wat.edu.pl
\* Correspondence: rafal.przesmycki@wat.edu.pl; Tel.: +48-504-059-739

**Abstract:** The field of wireless cellular network technology has seen a significant development in recent years, allowing the emergence of many new applications in addition to the traditional mobile phone calls. We are currently implementing the 5G system, which is replacing the previous cellular technologies on the market. Parallel to the development of cellular technologies, wireless local networks based on the IEEE 802.11× standards are rapidly spreading. The desire to use the advantages of both mobile telephony and wireless local networks has led to the idea of integrating the currently used communication systems in one device and requires a well-designed antenna, which should be given a lot of attention when designing the radio system. This article presents the proposed model of a two-band microstrip antenna for which the main assumption is its operating frequencies in the LTE-U (LTE-Unlicensed) band and one of the 5G system bands. The antenna dimensions and parameters have been calculated, simulated, and optimized using CST Microwave Studio software. The developed antenna has a compact structure with dimensions of ($60 \times 40 \times 1.57$) mm. The dielectric material RT Duroid 5880 with a dielectric constant $\varepsilon_r$ = 2.2 and thickness $h$ = 1.57 mm was used as a substrate for the antenna construction. The article presents an analysis of the results of simulation and measurements of selected electrical parameters and radiation characteristics of the proposed antenna. The antenna described in the article, working in 5G systems and LTE-U systems, is characterized by two operating bands with center frequencies equal to 3.52 GHz and 5.37 GHz, a low reflection coefficient (for resonances −31.54 dB and −23.16 dB), a gain value of 4.45 dBi, a wide frequency band of 3.0 GHz (68.18%), and a high energy efficiency in the range of 80–96.68%.

**Keywords:** antenna; microstrip antenna; 5G antenna; LTE-U antenna; crescent antenna; 5G system; LTE-U system

## 1. Introduction

Radio communication and wireless technologies are an integral part of the functioning of today's societies. Everyday communication, from calling home from a mobile phone to advanced medical applications such as monitoring and diagnostic solutions, all use wireless technologies for communication.

The field of wireless cellular network technology has seen a significant development in recent years, allowing the emergence of many new applications in addition to the traditional mobile phone calls. The 3G and 4G data transmission standards have transformed cellular telephony and available wireless services. We are currently at the stage of implementing the 5G system, which is replacing the earlier technologies on the market. At the same time, the development of cellular technologies and wireless local area networks (WLAN), operating on the basis of the IEEE 802.11× (Wi-Fi) standards, are rapidly spreading. Currently, these systems offer bitrates much higher than cellular telephone systems. At the same time, the number of access points (hotspots) that offer the ability to connect to the network is constantly growing. The desire to use the advantages of both mobile telephony and wireless local networks led to the idea of integrating the currently used communication systems in one device. The future of telecommunications will belong to solutions that can flexibly adapt to changing technical and geographical conditions [1,2].

These systems are commonly used in cell phones that must support all of these radio systems simultaneously. A mobile telephone or smartphone must ensure that all radio systems that it supports are compatible. Currently, we use the 3G system on our phone or smartphone, which is called the "third-generation cellular" system. This standard enables the use of multimedia, video calls, and mobile internet. Another system used in smartphones is the 4G LTE (long-term evolution) system. This is the first significant improvement in the area of mobile telecommunications networks. The LTE standard ensured broadband internet access, fast data transfer, and a wide range of mobile services, including watching movies in HD quality. The 5G system is increasingly used in smartphones. It is the fifth generation of cellular networks, which is a significant step forward compared to the currently used 4G LTE networks. The 5G system enables data transfer speeds down to Gbps and offers a significantly greater bandwidth and short connection delays. The 5G networks are designed keeping in mind the rapid growth of data volumes and communication in modern societies, the Internet of Things, with billions of interconnected devices, and future innovations [3–5].

Mobile networks send and receive low-intensity radio signals. They are sent and received by antennas linked to radio devices (transmitters and receivers) known as cellular base transceiver stations. The base stations are connected to other mobile and fixed networks to which they transmit signals/calls. The dissemination of 5G networks requires the preparation of antenna infrastructure and the implementation of new technological solutions. A significant number of antennas (apart from mobile devices) will be installed inside buildings, especially inside public utility buildings, including stadiums, railway stations, and shopping centers. This fact shows how important an element a well-designed antenna is in radio systems, and therefore it should be given a lot of attention during designing a radio system [1,2].

This article presents the proposed model of a two-band microstrip antenna made in the CST Microwave Studio software, for which the main assumption is its operating frequencies in the LTE-U band and one of the 5G system bands. Other important assumptions for the antenna model are its small size (not to be larger than 60 mm × 60 mm) and elliptical polarization. In the antenna design process, optimization was made in terms of minimizing dimensions and minimizing weight, which will allow the use of the designed antenna in portable terminals and easy integration with electronic devices.

## 2. Characteristics of the 5G System

Until now, the implementation of next generations of mobile networks has involved, inter alia, the use of new radio techniques or the addition of new network elements. However, the roll-out of 3G was not dependent on whether the operator was already providing 2G services or not. These technologies work well together, but are functionally independent of each other. It is a bit different with 5G systems. The operation of this technology is functionally strongly related to the LTE (4G) network. The signal range that can be achieved using higher frequencies that are intended for 5G is important, such as the so-called C-Band, i.e., the 3400–3800 MHz range, or the mmWave (millimeter wave), i.e., 26–28 GHz [2,6–8].

The 3.6 GHz band allows the use of MIMO (multi-input multi-output), and at the same time is a compromise between propagation and capacity resulting from spectral resources, especially in combination with the 700 MHz band improving uplink. This band would be used to build a cover layer for eMBB (enhanced mobile broadband) services for several of the largest Polish cities, including communication routes between them. This band can also be used to introduce services requiring reliable transmission and a particularly low latency (URLLC—ultra reliable low latency communications) in applications requiring the transmission of particularly large amounts of data, e.g., high-definition images for medical or navigation purposes.

The 28 GHz band has a limited area of use, especially due to also meeting the requirements for transmission from the user to the base station ("uplink"). It can be used

for broadband internet hot spots and pico cell cMTC (massive machine-type communications)/URLLC applications. Due to its large capacity and the possibility of allocating large spectrum resources, this band can also be used to provide internet access as part of the fixed wireless access service.

An important conclusion resulting from the assumptions for the 5G system is that the reduction of propagation losses at the 3.6 GHz and 28 GHz frequencies takes place mainly in the base station thanks to the higher antenna gain and other techniques. Consequently, for 3.6 GHz (compared to 1.8 GHz) the downlink will have better coverage than the reverse uplink. The phone has limited dimensions and limited power, so it cannot use the same optimization procedures as the base station [9,10].

To sum up, the implementation of the fifth-generation network in the NSA (nonstandalone) model requires close cooperation between 4G and 5G technologies. An important element of cooperation is the fact that 4G uses lower frequency bands with better propagation properties. In the SA (standalone) model, the higher frequency bands also need the range support of the lower frequencies—with the difference that 5G technology is also activated on the lower band [4,5,11].

### 3. Characteristics of the LTE-U System

The LTE-U (LTE-Unlicensed) is a proposal, originally developed by Qualcomm, to use 4G LTE radio technology in an unlicensed spectrum such as the 5 GHz band used by 802.11a compliant Wi-Fi equipment. It will be an alternative to the operator's Wi-Fi hotspots. LTE networks carry more and more data. Cells can be made smaller to do this, but this is not a complete solution as more spectrum is needed [12,13].

One approach is to use an unlicensed spectrum along with licensed bands. Known in 3GPP as LTE-LAA (LTE license-assisted access) or more generally as LTE-U (LTE-Unlicensed), it allows access to an unlicensed spectrum, particularly in the 5 GHz ISM band.

A significant amount of unlicensed spectrum is available worldwide. These bands are used worldwide to provide unlicensed access for short-range radio transmissions. These bands, called ISM (industrial, scientific, and medical) bands, are allocated in different parts of the spectrum and are used for many different applications, including microwave ovens, Wi-Fi, Bluetooth, and many more [14,15].

The most interesting frequency band for LTE-U/LTE-LAA (license-assisted access) is the 5 GHz band. In this case, there are several hundred MHz of spectrum bandwidth available, although the exact bandwidth availability varies from country to country [12]. The division of the 5 GHz band used for LTE-U is shown in Figure 1.

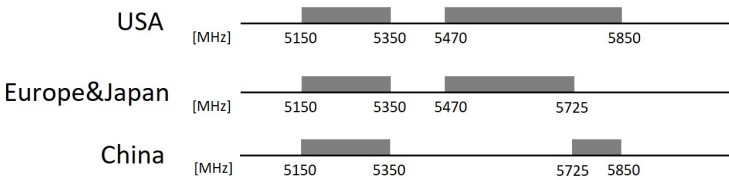

**Figure 1.** LTE-U frequency band.

In addition to the basic frequency limits, the use of the 5 GHz bands for applications such as LTE-U or LTE-LAA has some regulatory requirements.

One of the main requirements for accessing these frequencies is the ability to coexist with other users of the band—the CCA (clear channel assessment) or LBT (listen-before-talk) method is required. This often means that instant access may not always be available when LTE-U is deployed. Another important limit is that different power levels are allowed depending on the country and area of the frequency band used. Typically, between 5150 and 5350 MHz, the maximum power limit is 200 mW and operation is limited to indoor use only, and higher frequencies often allow power levels up to 1 W [14,15].

The use of LTE-U/LTE-LAA was first introduced in the Rel13 3GPP standard. Basically, LTE-U is based on the LTE-Advanced carrier aggregation capability, which has been

implemented since around 2013. Carrier aggregation aims to increase the overall capacity available to the user equipment by allowing it to use more than one channel on the same frequency band or on a different band [12].

There are several ways to implement LTE-U:

–　Downlink only: This is the most basic form of LTE-U and is similar to some of the first LTE carrier aggregation implementations. In this case, the primary cell link is always within the licensed spectrum bands (Figure 2).

–　Uplink and downlink: Full operation of TDD LTE-U with user equipment having an uplink and downlink connection in an unlicensed spectrum requires more functions to be enabled (Figure 3).

–　FDD/TDD Aggregation: LTE-CA allows a combination of carrier aggregation between FDD and TDD. This provides much more flexibility in selecting the band to be used in the unlicensed spectrum for LTE-LAA operation [16,17].

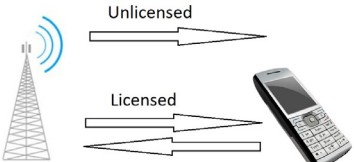

**Figure 2.** LTE-U downlink only.

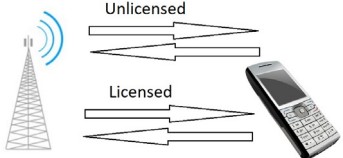

**Figure 3.** LTE-U uplink and downlink.

The LTE-U builds on the existing backbone for backhaul and other capabilities such as security and authentication. Therefore, no changes to the backbone are needed. Some changes to the base station are needed to adapt to the new frequencies, and to take into account the capabilities required to ensure proper sharing of unlicensed frequencies. In addition, phones or devices in the EU will need to have the new LTE-U/LTE-LAA feature built in to be able to access LTE on these additional frequencies [12].

The LTE-U would enable mobile operators to increase the coverage of their mobile networks, using the unlicensed 5 GHz band, in which Wi-Fi devices already operate. T-Mobile and Verizon Wireless showed interest in implementing such a system as early as 2016. While cellular network providers tend to rely on radio spectrum for which they have exclusive licenses, LTE-U would share space with Wi-Fi devices already existing there (smartphones, laptops, and tablets connecting to home broadband networks, free hotspots provided by companies, etc.).

The LTE-U is designed to enable cellular networks to increase data transmission speeds over short distances without having to use a separate Wi-Fi network. Unlike Wi-Fi connections, there is a control channel using LTE, but all data (not only phone calls) flow through the unlicensed 5 GHz band [12].

## 4. Analysis Dual-Band Antenna Solutions Operating in LTE and 5G Systems at 3.6 GHz

In the literature, there is a small number of proposed solutions for dual-band microstrip antennas operating only in 5G systems at the frequency of 3.6 GHz and simultaneously in the 5 GHz band [18–37]. The published solutions are characterized by a compact solution, small geometric dimensions, and a wide bandwidth. Many of these solutions are designed on the RT Duroid 5880 laminate with a dielectric constant $\varepsilon_r$ equal to 2.2, and on the FR 4

laminate with a dielectric constant $\varepsilon_r$ equal to 4.2. In the analysed antennas, the radiating patch in most cases has a rectangular shape and is powered by a microstrip line. The differences in the proposed antennas concern the modification of the radiator shape on the basis of solutions used in fractal antennas. As a result, different bandwidths of the proposed antennas are obtained.

The antenna proposed in the article, working in LTE-U and 5G systems, is designed for two resonant frequencies, equal to 3.6 GHz and 5.5 GHz, respectively, using the RT Duroid 5880 laminate and a crescent patch. The proposed antenna is designed to achieve a bandwidth covering the frequency ranges from 3.4 GHz to 3.8 GHz (operating frequencies of the 5G system) and from 5.150 GHz to 5.850 GHz (operating frequencies of the LTE-U system).

## 5. Dual-Band Microstrip Antenna Designed for LTE-U and 5G Systems

Before developing an appropriate numerical model of the designed antenna in the simulation environment, it is necessary to perform preliminary calculations of its geometric dimensions based on the parameters of the dielectric substrate and the resonance frequencies of the antenna. These activities are aimed at obtaining a preliminary model of the antenna, which will ensure the compliance of the structure with the assumptions made for it and improve and shorten the achievement of the assumed target in the process of the simulation and optimization of the antenna structure.

The main assumption for the designed microstrip antenna operating in 5G and LTE-U systems is the frequency range, which should cover the frequency bands from 3.4 GHz to 3.8 GHz (5G system operating frequencies) and from 5.150 GHz to 5.850 GHz (LTE-U system operating frequencies). In addition to the frequency band, another important requirement for the designed antenna are the dimensions of the antenna, which should not be larger than 60 mm × 60 mm, and the omnidirectional radiation pattern with elliptical polarization ensuring the reception of signals with all polarities. The main parameter on which the dimensions of the antenna will depend is its resonant frequency $f_r$ and the relative electric permittivity $\varepsilon_r$ of the dielectric layer of the substrate on which it will be made. The thickness of the substrate directly affects the efficiency and bandwidth of the microstrip antenna. As the thickness of the substrate increases, the antenna operating bandwidth increases and its efficiency decreases.

One of the methods of obtaining elliptical polarization for the numerical model of the designed microstrip antenna is to use a patch in the form of a circle. The process of determining the parameters of a microstrip antenna with a circular patch is very similar to that of an antenna with a rectangular patch. In order to determine the diameter of the patch, use the following dependencies [19,38–40]:

$$
\begin{aligned}
2R &= \frac{F}{\sqrt{\left\{1 + \frac{2h}{\pi \varepsilon_r F}\left[ln\left(\frac{\pi F}{2h}\right) + 1.7726\right]\right\}}} = 12.88 \text{ mm, for } f_r = 3.6 \text{ GHz}\\
2R &= \frac{F}{\sqrt{\left\{1 + \frac{2h}{\pi \varepsilon_r F}\left[ln\left(\frac{\pi F}{2h}\right) + 1.7726\right]\right\}}} = 7.79 \text{ mm, for } f_r = 5.5 \text{ GHz}
\end{aligned}
\tag{1}
$$

where $2R$ is the diameter, $R$ is the radius, $h$ is the laminate thickness, and $\varepsilon_r$ is the dielectric constant.

$$
\begin{aligned}
F &= \frac{1.8412 \cdot 10^9}{f_r \sqrt{\varepsilon_r}} = 16.46 \text{ mm, for } f_r = 3.6 \text{ GHz}\\
F &= \frac{1.8412 \cdot 10^9}{f_r \sqrt{\varepsilon_r}} = 10.77 \text{ mm, for } f_r = 5.5 \text{ GHz}
\end{aligned}
\tag{2}
$$

Based on the above values, in the next step we determine the effective radius from the relationship [19,38–40]:

$$
\begin{aligned}
2R_e &= 2R \cdot \sqrt{\left\{1 + \frac{2h}{\pi \varepsilon_r 2R}\left[ln\left(\frac{\pi 2R}{2h}\right) + 1.7726\right]\right\}} = 17.19 \text{ mm, for } f_r = 3.6 \text{ GHz}\\
2R_e &= 2R \cdot \sqrt{\left\{1 + \frac{2h}{\pi \varepsilon_r 2R}\left[ln\left(\frac{\pi 2R}{2h}\right) + 1.7726\right]\right\}} = 11.56 \text{ mm, for } f_r = 5.5 \text{ GHz}
\end{aligned}
\tag{3}
$$

The last element of the antenna design is to determine the dimensions of the feed line. The calculation of the dimensions of the microstrip feed line with the characteristic impedance $Z_C = 50 \, \Omega$ begins with the determination of the relationship of auxiliary variables $a$ and $b$ [19,38–40]:

$$a = \frac{Z_c}{60} \sqrt{\frac{\varepsilon_r + 1}{2}} + \frac{\varepsilon_r - 1}{\varepsilon_r + 1} \left( 0.23 + \frac{0.11}{\varepsilon_r} \right) = 1.16$$
$$b = \frac{60 \pi^2}{Z_c \sqrt{\varepsilon_r}} = 7.98 \tag{4}$$

Since the parameter $a$ is less than 1.52, the width and length of the feed line are determined from the following equation [19,38–40]:

$$W_f = A = \frac{2}{\pi} \left\{ b - 1 - \ln(2b - 1) + \frac{\varepsilon_r - 1}{2\varepsilon_r} \left[ \ln(b - 1) + 0.39 - \frac{0.61}{\varepsilon_r} \right] \right\} * h = 6.99 \text{ mm}$$
$$L_f = 3 \cdot h = 3 \cdot 1.57 = 4.71 \text{ mm} \tag{5}$$

The determination of the length of the feed line completes the stage of determining the input data for the initial simulation model. Based on the calculations made, assuming the basic data for the 3.6 GHz frequency, a model was obtained, the preliminary dimensions of which are shown in Table 1. Unfortunately, in this model, for the frequency equal to 5.5 GHz, no resonance was obtained, and therefore it was decided to correct the calculations using the optimization process available in the CST Microwave Studio software. Thanks to this process, the final version of the antenna was obtained after modifications were made to the following: the shape of the radiating patch, by making a circular indentation in the circular patch with a diameter resulting from the radiator size for f = 5.5 GHz; the dimensions of the feed line; and the size of the reference plane (antenna shield). The final dimensions of this antenna are presented in Table 1. The appearance of the final version of the antenna model with a crescent radiating patch is shown in Figure 4, while Figure 5 shows the appearance of the physical antenna model [41,42].

**Table 1.** Preliminary and final dimensions of the proposed antenna.

| Layer | | Preliminary Dimensions [mm] | Final Dimensions [mm] |
|---|---|---|---|
| Ground plane width | $W_e$ | 42.36 | 40.00 |
| Substrate width | $W_s$ | 42.36 | 40.00 |
| Substrate length | $L_s$ | 36.67 | 60.00 |
| Ground plane length | $L_e$ | 36.67 | 39.60 |
| Patch diameter | $2R_1$ | 17.19 | 18.70 |
| Patch radius | $R_1$ | 8.59 | 9.35 |
| Patch cut diameter | $2R_2$ | 11.56 | 13.02 |
| Patch cut radius | $R_2$ | 5.78 | 6.51 |
| Copper thickness | Cu | 0.035 | 0.035 |
| Substrate thickness | h | 1.57 | 1.57 |
| Permittivity | $\varepsilon_r$ | 2.2 | 2.2 |
| Feed line width | $W_f$ | 6.99 | 2.80 |
| Feed line length | $L_f$ | 4.71 | 33.50 |

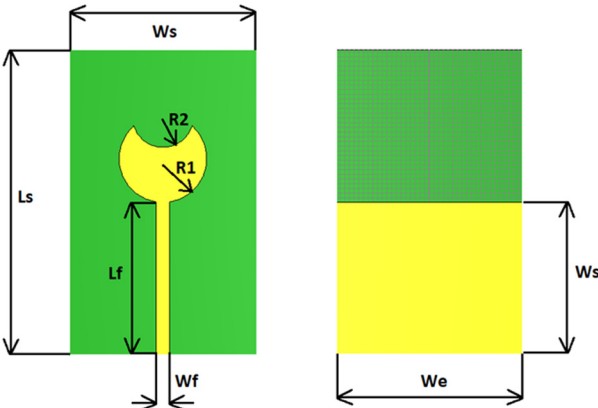

**Figure 4.** The optimized antenna model view—front and back side with dimensions.

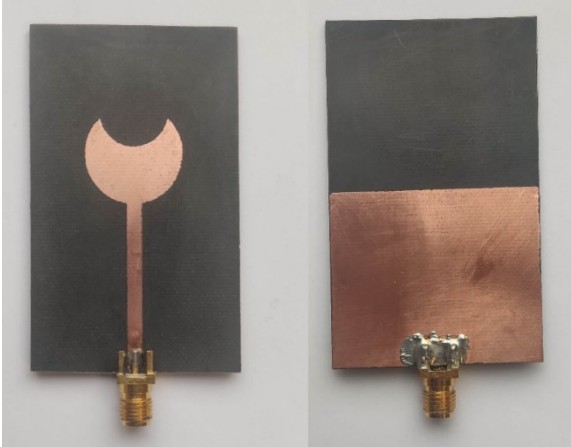

**Figure 5.** The optimized physical antenna model view—front and back side with dimensions.

## 6. Simulations and Measurements Results

The designed antenna structure consists of three components: a ground plane, patch, and substrate. The analysis of the electrical parameters of the radiating element and other elements of the preliminary antenna model showed that it is possible to improve the electrical parameters of the antenna, such as reducing VSWR, increasing the bandwidth, miniaturization of the antenna dimensions, or increasing the energy gain. For this purpose, the optimization process was carried out in the CST Microwave Studio software in terms of the above-mentioned parameters. For the calculated parameters of the patch, a preliminary simulation of the electrical parameters of the developed antenna model was performed and the process of optimization of the structure was carried out, assuming that the main assumptions for the antenna remained unchanged [41,42].

For the final model of the antenna designed in this way, a simulation process was carried out using the CST Microwave Studio software, thus obtaining the results of the electrical parameters, such as the reflection coefficient, voltage standing wave ratio, input impedance, energy gain, antenna efficiency, antenna current distribution, and radiation patterns. Additionally, selected electrical parameters were measured for the physical model of the antenna. The appearance of the proposed antenna during the measurements is shown in Figure 6, during the measurements of electrical parameters and the measurement of radiation characteristics, respectively.

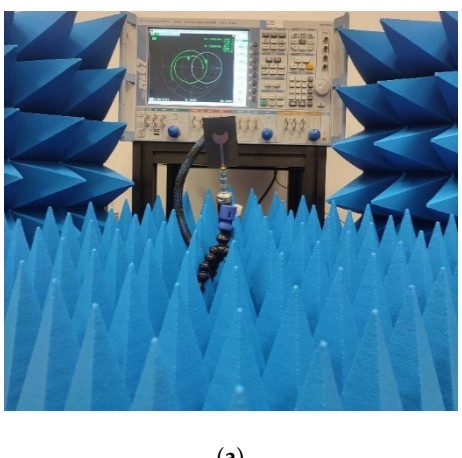
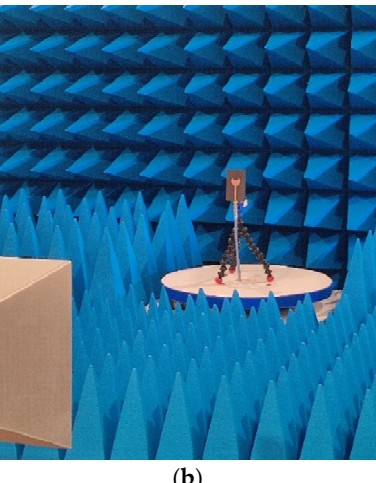

(**a**)              (**b**)

**Figure 6.** Appearance of the laboratory stand during the measurements of (**a**) electrical parameters of the antenna and (**b**) radiation patterns of the antenna.

### 6.1. Reflection Coefficient $S_{11}$

The base reflectance value was taken to be $-10$ dB, which means that 10% of the incident power is reflected, i.e., 90% of the power is received by the antenna, which is considered good for mobile communication. Figure 7 shows the results of the reflection coefficient as a function of frequency for the proposed antenna. The continuous line shows the results of the simulations obtained in CST Microwave Studio, while the dashed line shows the results of the measurements made for the physical model of the antenna. The proposed antenna has two resonances at 3.75 GHz with return losses of $-13.70$ dB and 5.17 GHz with return losses of $-20.17$ dB for the simulation results, and at 3.52 GHz with a reflection coefficient of $-31.54$ dB and 5.42 GHz with a reflection coefficient of $-23.15$ dB for the measurement results. The antenna has a bandwidth of 3.00 GHz, which gives a relative bandwidth of 68.18%.

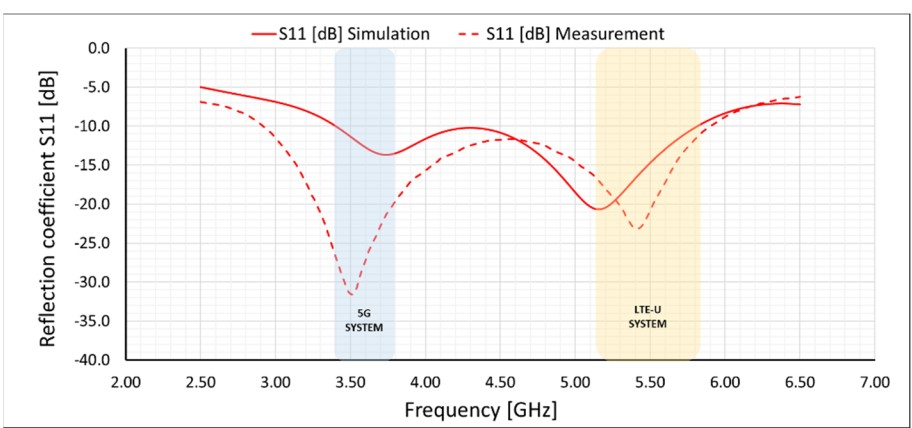

**Figure 7.** The reflection coefficient as a function of frequency for the proposed antenna working in 5G and LTE-U systems.

### 6.2. Voltage Standing Wave Ratio

For a microstrip antenna, the voltage standing wave ratio (VSWR) should not be greater than 2 across the entire frequency bandwidth. Ideally, this value should be 1. Figure 8 shows the voltage standing wave ratio as a function of frequency for the proposed antenna. The continuous line shows the results of the simulations obtained in CST Microwave Studio, while the dashed line shows the results of the measurements made for the physical model of the antenna. The VSWR value obtained for the simulation results at 3.75 GHz resonance frequency was 1.52, and at 5.17 GHz it was 1.20. The VSWR value

obtained for the measurement results at a resonance frequency of 3.52 GHz was 1.07, and at the frequency of 5.42 GHz it was 1.18. The values presented in Figure 8 show that the proposed antenna works in the entire assumed frequency band (for the 5G and LTE-U system), i.e., from 3.40 GHz to 5.85 GHz.

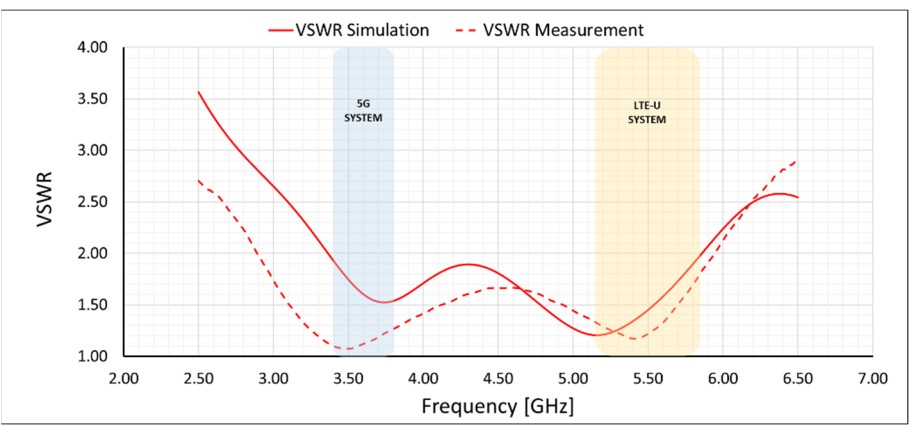

**Figure 8.** The voltage stand wave ratio as a function of frequency for the proposed antenna working in 5G and LTE-U systems.

### 6.3. Input Impedance

The antenna design assumes that the impedance of the feed line should be 50 Ω. In the case of large discrepancies, it is possible to use a matching system, but it is another system that introduces additional losses, and in financial terms generates additional costs. The input impedance as a function of frequency for the proposed antenna is shown in Figure 9. The continuous lines show the results of the simulations obtained in CST Microwave Studio, while the dashed lines show the results of the measurements made for the physical model of the antenna.

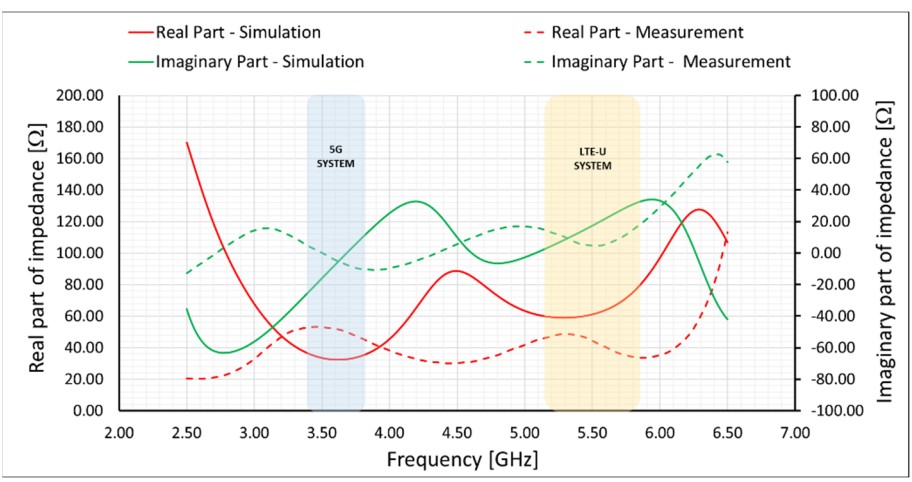

**Figure 9.** The input impedance as a function of frequency for the proposed antenna working in 5G and LTE-U systems (real part—red line, imaginary part—green line).

### 6.4. Antenna Gain

Most often, the antenna gain is given in relation to the isotropic antenna and is expressed in dBi units. Sometimes it is also given in relation to a dipole antenna and is expressed in units of dBd. The antenna gain depends on its directivity and antenna energy losses are dependent on the material from which it is made. The value of the antenna gain as a function of frequency is shown in Figure 10. The continuous line shows the results of the simulations obtained in CST Microwave Studio, while the dashed line shows the results

of the measurements made for the physical model of the antenna. The proposed antenna has a maximum energy gain of 4.51 dBi at the resonance frequency of 3.70 GHz for the simulation results and 4.35 dBi at the resonance frequency of 3.70 GHz for the measurement results.

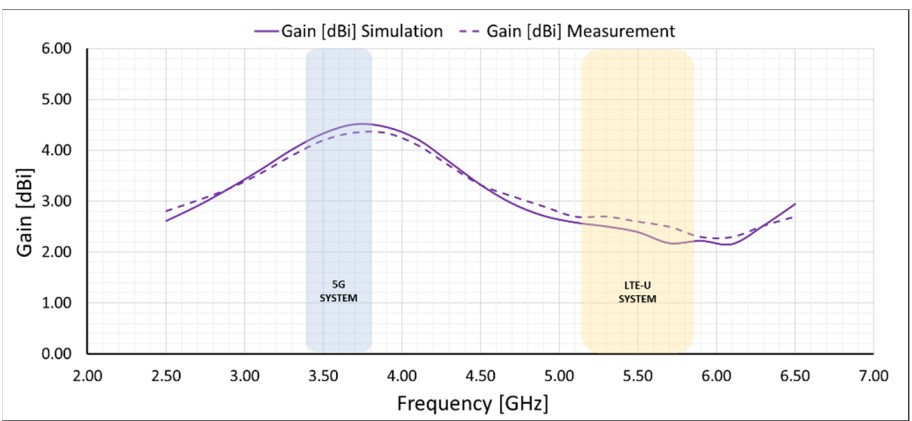

**Figure 10.** The antenna gain as a function of frequency for the proposed antenna working in 5G and LTE-U systems.

### 6.5. Efficiency

Antenna efficiency is the term used to describe the relationship between the amount of radiated power and the power delivered to the antenna. Antenna efficiency often helps identify any problems with the antenna design itself, and also helps identify other factors that may interfere with the antenna's ability to receive signals efficiently. For the proposed antenna, only the simulation process was performed to determine the antenna efficiency as a function of frequency. The efficiency value of the proposed antenna as a function of frequency is shown in Figure 11. The proposed antenna has a high energy efficiency in the range of 80–96.68%.

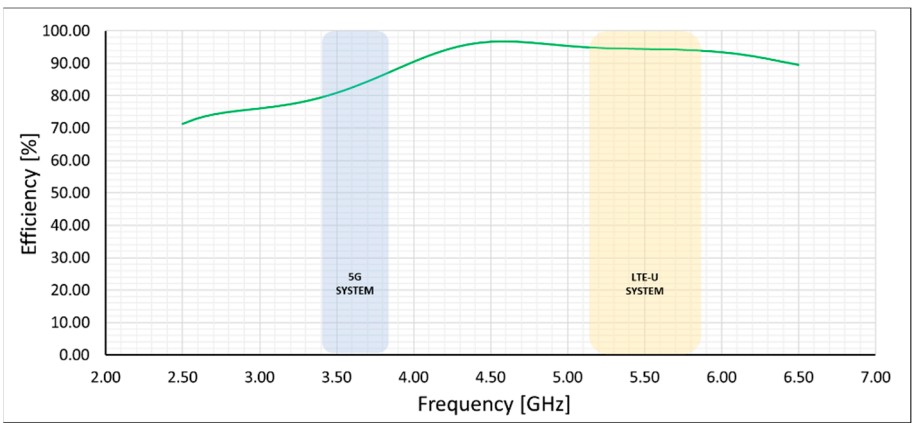

**Figure 11.** The antenna efficiency as a function of frequency for the proposed antenna working in 5G and LTE-U systems.

### 6.6. Current Distribution in the Antenna

In the microstrip antenna at the end of the radiating element (edge of the patch), the current value should be minimal. The voltage at the edge of the patch is out of phase with the current. Consequently, the voltage will peak at the tip of the patch with currents close to zero. The voltage out of phase with the current phase creates fields at the edges of the microstrip antenna. Figure 12 shows the current distribution of the proposed antenna for the frequencies of 3.6 GHz and 5.5 GHz (resonant frequencies).

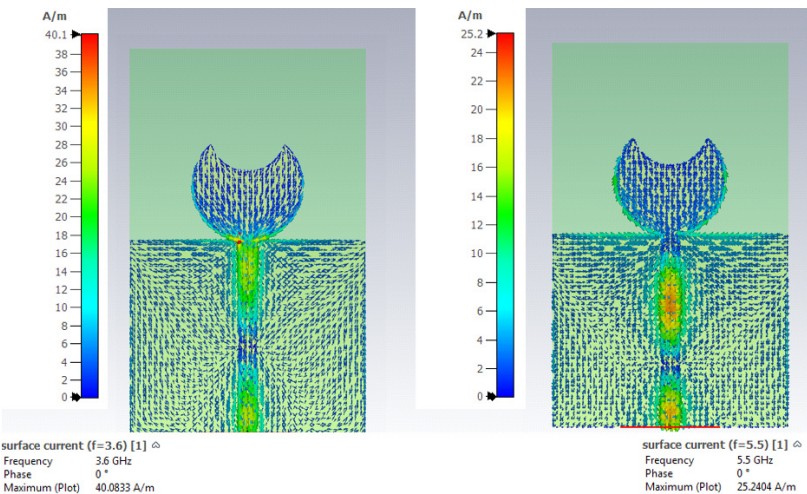

**Figure 12.** Surface current distribution for the proposed 5G and LTE-U antenna at 3.6 GHz and 5.5 GHz frequency.

### 6.7. Radiation Patterns

The radiation pattern shows how the antenna radiates energy depending on the direction. It represents the normalized distribution of the electric field or the relative distribution of the surface power density. The radiation patterns are determined in two planes, horizontal and vertical, and can also be presented in a three-dimensional form. The designed antenna should have an omnidirectional radiation pattern. Figures 13 and 14 show the three-dimensional appearance of the radiation pattern of the proposed antenna for the assumed center frequencies of the 5G system and the LTE system, 3.6 GHz and 5.5 GHz, respectively. Figure 15 shows the normalized radiation patterns of the proposed antenna for the frequency of 3.6 GHz (black line simulation, red line measurement) in the polar coordinate system for vertical and horizontal polarization planes. Figure 16 shows the normalized radiation patterns of the proposed antenna for the 5.5 GHz frequency (black line simulation, red line measurement) in the polar coordinate system for vertical and horizontal polarization planes.

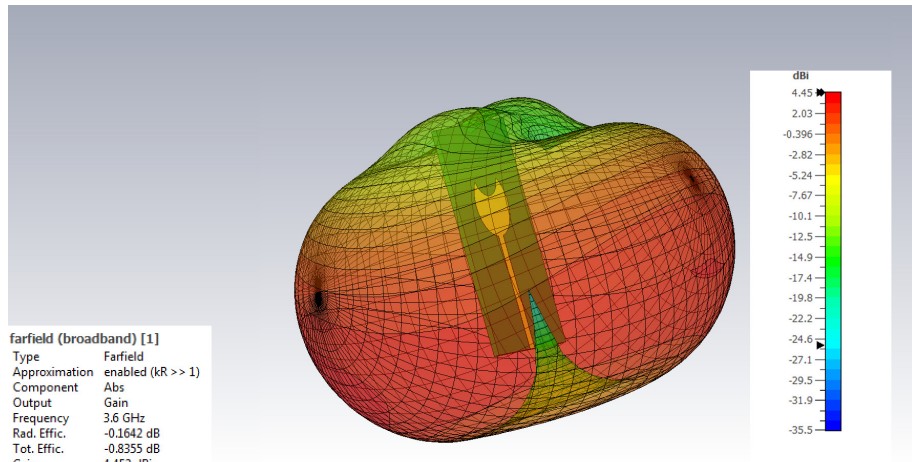

**Figure 13.** The 3D view of the radiation pattern for the proposed antenna model at 3.60 GHz frequency.

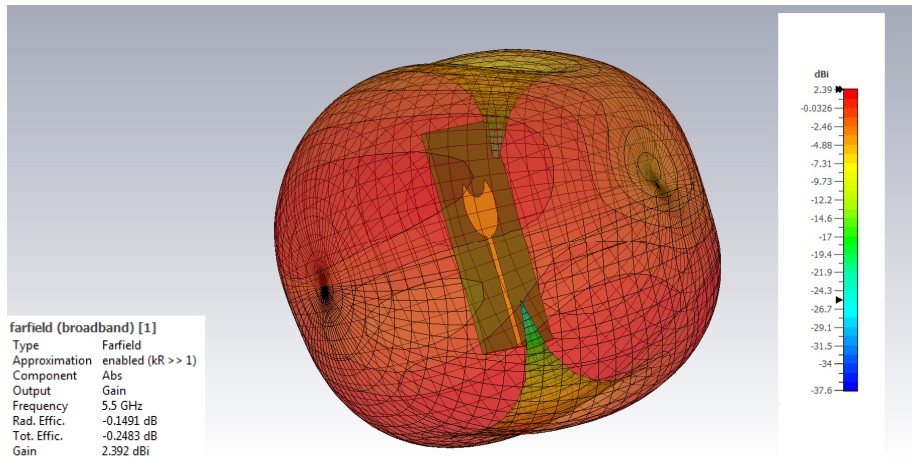

**Figure 14.** The 3D view of the radiation pattern for the proposed antenna model at 5.50 GHz frequency.

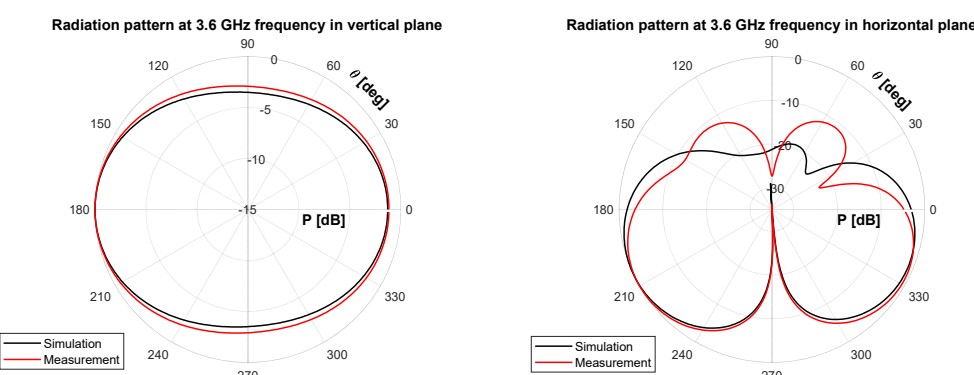

**Figure 15.** The normalized radiation patterns for the proposed antenna model operating in 5G and LTE-U systems for 3.6 GHz (black line—simulation results, red line—measurement results) in polar coordinates for vertical and horizontal planes.

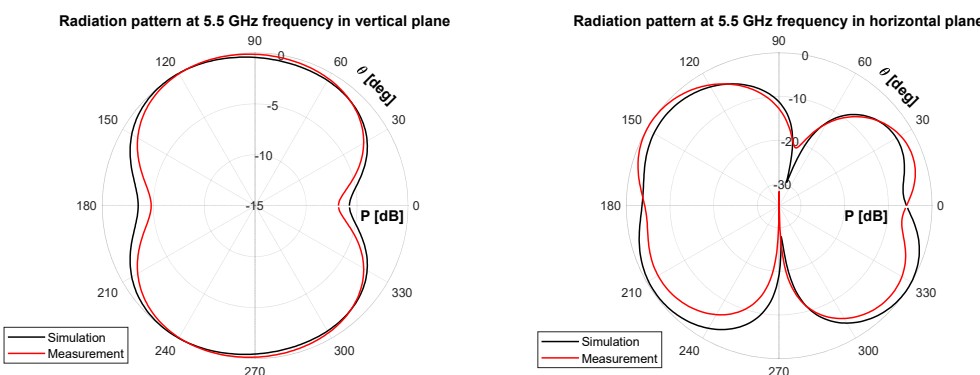

**Figure 16.** The normalized radiation patterns for the proposed antenna model operating in 5G and LTE-U systems for 5.5 GHz (black line—simulation results, red line—measurement results) in polar coordinates for vertical and horizontal planes.

## 7. Comparison of the Proposed Antenna with Other Antennas

The parameter values for the proposed antenna in terms of impedance matching and bandwidth can be compared with other published results in order to obtain a comparative assessment. The frequency response of the measured parameter $S_{11}$ for the proposed antenna is not the lowest (especially for comparing the simulation value) compared to the

obtained value of $S_{11}$ of the antennas presented in [22–24,27,34], but it is relatively low. Table 2 shows a comparison of the electrical parameters of the proposed antenna with other selected antennas available in the literature. The table shows a comparison of the reflection coefficient, the frequency bandwidth, and the gain of the antennas.

**Table 2.** Comparison of electrical parameters of the proposed antenna with other antennas, taking into account the bandwidth.

| Performance Measure | | Proposed Antenna Measurement | Work of [22] Simulation | Work of [23] Simulation | Work of [24] Simulation | Work of [27] Simulation | Work of [34] Simulation |
|---|---|---|---|---|---|---|---|
| Center frequency | | 4.40 GHz | NA | NA | NA | 7.50 GHz | NA |
| BW | $S_{11} \leq -10.00$ dB | 3.00 GHz | NA | NA | NA | 9.00 GHz | NA |
| Center frequency no 1 | | 3.52 GHz | 3.50 GHz | 3.60 GHz | 3.60 GHz | NA | 3.8 GHz |
| | $S_{11} \leq -10.00$ dB | 3.00 GHz | 0.20 GHz | 0.35 GHz | 0.20 GHz | NA | 0.15 GHz |
| BW | $S_{11} \leq -13.97$ dB | 1.02 GHz | 0.15 GHz | 0.30 GHz | NA | NA | 0.10 GHz |
| | $S_{11} \leq -19.08$ dB | 0.55 GHz | 0.10 GHz | 0.20 GHz | NA | NA | NA |
| Center frequency no 2 | | 5.37 GHz | 5.50 GHz | 5.45 GHz | 5.50 GHz | NA | 5.25 GHz |
| | $S_{11} \leq -10.00$ dB | 3.00 GHz | 0.25 GHz | 0.60 GHz | 0.55 GHz | NA | 0.55 GHz |
| BW | $S_{11} \leq -13.97$ dB | 0.70 GHz | 0.15 GHz | 0.45 GHz | 0.40 GHz | NA | 0.45 GHz |
| | $S_{11} \leq -19.08$ dB | 0.30 GHz | NA | 0.30 GHz | 0.33 GHz | NA | 0.20 GHz |
| Relative BW ($S_{11} \leq -10.00$ dB) | | 68.18% | NA | NA | NA | 120% | NA |
| Relative BW no 1 ($S_{11} \leq -10.00$ dB) | | 68.18% | 5.71% | 9.72% | 5.55% | 120% | 3.94% |
| Relative BW no 2 ($S_{11} \leq -10.00$ dB) | | 68.18% | 4.54% | 11.00% | 10.00% | 120% | 10.40% |
| Max Gain | | 4.35 dBi | 3.10 dBi | NA | 3.00 dBi | NA | 5.8 dBi |

Based on comparisons of the electrical parameters of the proposed antenna with other antennas, it has been shown that the proposed dual-band antenna has the comparable performance in terms of impedance matching in all cases, especially in cases of stringent matching conditions ($S_{11} \leq -10$ dB, $S_{11} \leq -13.97$ dB and $S_{11} \leq -19.08$ dB) and the values of antenna gain.

## 8. Conclusions

Due to the growing demand for mobile data and mobile devices, for 5G applications and applications using broadband LTE internet access, this article proposes a dual-band crescent microstrip antenna. The proposed antenna has two resonant frequencies, the first at 3.52 GHz with a reflection coefficient of $-31.54$ dB and the second at 5.42 GHz with a reflection coefficient of $-23.15$ dB. The proposed antenna covers the frequency ranges from 3.4 GHz to 3.8 GHz (operating frequencies of the 5G system) and from 5.150 GHz to 5.850 GHz (operating frequencies of the LTE-U system). The proposed antenna shows efficiency in the range of 80.00–96.68%, and the maximum antenna gain for the resonance frequency of 3.70 GHz is 4.35 dBi. The results also show that its bandwidth is 3.00 GHz (relative bandwidth: 68.18%), which is a very good result, much greater (but not the best) than the results of other works published in the world, e.g., [22–24,34], where the operating band of the proposed of antennas is of the order of 0.6 GHz (11.00%). The proposed antenna can serve as a good option for 5G mobile communication and wireless access to local networks that require high bandwidth. The size of the antenna is very compact and its weight is very low, making it suitable for devices where space is the main limitation.

**Author Contributions:** Conceptualization, R.P.; methodology, M.B. and R.P.; resources, M.B.; validation, M.B.; visualization, R.P. and M.B.; writing—original draft preparation, R.P. and M.B; writing—review and editing, R.P. All authors have read and agreed to the published version of the manuscript.

**Funding:** This work was financed by Military University of Technology under research project UGB-22-741/2022 on "Electrical parameters measurement of solid materials using the reflection and the free space method in the millimeter band".

**Conflicts of Interest:** The authors declare no conflict of interest.

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
