# Peer review of "Crescent Microstrip Antenna for LTE-U and 5G Systems"

_electronics, doi:10.3390/electronics11081201_

Round 1

Reviewer 1 Report

The authors failed to properly address the comments for the previous version of the paper and to significantly  improve it.

Former review comments:

This type of layout has been proposed and analyzed before. For example [A. K. H. Obsiye, H. E. AbdEl-Raouf and R. El-Islam, "Modified printed crescent patch antenna for Ultrawideband RFID (UWB-RFID) tag," 2008 IEEE International RF and Microwave Conference, 2008, pp. 274-276, doi: 10.1109/RFM.2008.4897454.] shows the same geometry and frequency range. The improvements proposed by the authors are based on layout optimization using commercial electromagnetic simulation software (CST). Even so, the results are not better than those published in 2008 (bandwidth between 3.2-12 GHz, smaller size of 30 x 35 mm, etc.).   

Out of the thousands of microstrip patch antennas that could have been used for performance comparison, the three papers chosen in Table 2 all deal with complex multi-element antenna systems and thus should not be compared with the performance of a single element antenna.   

Reviewer 2 Report

This article presents the proposed model

of a two-band microstrip antenna made in the CST Microwave Studio software, for which the main

assumption is its operating frequencies in the LTE-U (LTE-Unlicensed) band and one of the 5G system bands. The antenna dimensions and parameters have been calculated, simulated and optimized

usining CST software.The paper is well written and the presentation is good hence

is recommended for pupublication in the Electronics. 

Reviewer 3 Report

I would recommend the acceptance of this article after removing the comparison between the measured data that are plotted by taking point by point from other works. The Table of comparison is enough to show how this work is different than other work,

Author Response

This manuscript is a resubmission of an earlier submission. The following is a list of the peer review reports and author responses from that submission.

Round 1

Reviewer 1 Report

The paper describes a single element modified circular patch antenna for LTE-U and sub-6GHz band 5G applications.

This type of layout has been proposed and analyzed before. For example [A. K. H. Obsiye, H. E. AbdEl-Raouf and R. El-Islam, "Modified printed crescent patch antenna for Ultrawideband RFID (UWB-RFID) tag," 2008 IEEE International RF and Microwave Conference, 2008, pp. 274-276, doi: 10.1109/RFM.2008.4897454.] shows the same geometry and frequency range. The improvements proposed by the authors are based on layout optimization using commercial electromagnetic simulation software (CST). Even so, the results are not better than those published in 2008 (bandwidth between 3.2-12 GHz, smaller size of 30 x 35 mm, etc.).   

Out of the thousands of microstrip patch antennas that could have been used for performance comparison, the three papers chosen in Table 2 all deal with complex multi-element antenna systems and thus should not be compared with the performance of a single element antenna.   

Reviewer 2 Report

This paper presents a two-band microstrip antenna for which the main 
assumption is its operating frequencies in the LTE-U (LTE-Unlicensed) band and one of the 5G system bands. The proposed antenna dimensions and parameters have been calculated, simulated, and optimized using CST Microwave Studio software.

The paper is well written and the presentation of the work is good.  However, the Figures' quality is poor which needs improvement in the revised version. 

Reviewer 3 Report

The proposed design is adopting modification of making circular cut in the basic layout of the circular patch antenna to achieve an acceptable matching level. However, similar performance can be achieved by using simple circular or rectangular patch with inset feeding matching. On the other hand, the authors mentioned that the idea of adding a patch in the shape of a circle to the designed microstrip antenna is to achieve elliptical polarization, however there is no evidence of this in the results.

Furthermore, feeding techniques affects bandwidth and cross-polarization levels. Microstrip feeding networks suffer from high ohmic, dielectric losses and surface waves. So, other feed mechanisms have been used to overcome these drawbacks. Therefore, a modern guiding structure such as printed ridge gap waveguide (PRGW) has been emerging. It is a promising guiding structure for sub-6 GHz applications providing low signal distortion and low losses compared to microstrip line.

Also, the authors mentioned that there is a small number of proposed solutions for dual-band microstrip antennas operating in 5G systems at the frequency of 3.6 GHz and simultaneously in the 5 GHz band. Moreover, it is noticeable after looking at the references mentioned that the comparison did not address similarities of the same work in terms of the form used in the same frequency bands. So, these are some of the attempts made by researchers that dealt with the topic in different ways.

Also, I am wondering how the authors compared measured results with other measured or simulated results of other works in the literature.

Here, some of the previous works use different techniques to obtain dual-band response of conventional patch antenna or for a wide band that combines the two bands.

Cheng, Y., & Dong, Y. “Wideband Circularly Polarized Split Patch Antenna Loaded With Suspended Rods”. IEEE Antennas and Wireless Propagation Letters, vol. 20, no. 2, pp. 229-233, 2020.‏

Singh, K. J., & Mishra, R. “A circular microstrip patch antenna with dual band notch characteristics for UWB applications”. In International Conference on Power Energy, Environment and Intelligent Control (PEEIC) pp. 153-156, April, 2018.‏

Azim, R., Meaze, A. M. H., Affandi, A., Alam, M. M., Aktar, R., Mia, M. S., ... & Islam, M. T. “A multi-slotted antenna for LTE/5G Sub-6 GHz wireless communication applications”. International Journal of Microwave and Wireless Technologies, vol. 13, no. 5, pp. 486-496, 2021.‏

Hsiao, C. W., & Chen, W. S. “Broadband dual-polarized base station antenna for LTE/5G C-band applications”. In Cross Strait Quad-Regional Radio Science and Wireless Technology Conference (CSQRWC), pp. 1-3, July, 2018.‏

Arifianto M. S., & Munir, A. “Dual-band circular patch antenna incorporated with split ring resonators metamaterials”. In 10th International Conference on Electrical Engineering/Electronics, Computer, Telecommunications and Information Technology, pp. 1-4, May, 2013.‏

Guo, X., Liao, W., Zhang, Q., & Chen, Y. “A dual-band embedded inverted T-slot circular microstrip patch antenna”. In IEEE 5th Asia-Pacific Conference on Antennas and Propagation (APCAP), pp. 151-152, July, 2016.‏

Dai, X. W., Zhou, T., & Cui, G. F. “Dual-band microstrip circular patch antenna with monopolar radiation pattern”. IEEE Antennas and Wireless Propagation Letters, vol. 15, pp. 1004-1007, 2015.

Veysi, M., Kamyab, M., & Jafargholi, A. “ Single-feed dual-band dual-linearly-polarized proximity-coupled patch antenna. IEEE Antennas and Propagation Magazine, vol. 53, no. 1, pp. 90-96, 2011.‏

And, other articles present a bandwidth-enhanced microstrip patch antenna with high gain.

Wen, J., Xie, D., & Zhu, L. “Bandwidth-enhanced high-gain microstrip patch antenna under TM50 dual-mode resonances”. IEEE Antennas and Wireless Propagation Letters, vol. 18, no. 10, pp. 1976-1980, 2019.‏

Gowda, M. N., Gajera, H. R., Poornima, S., & Sowjanya, N. B. “Dual Band Generation using Circular Ring Shaped Super Conducting Layer in Microstrip Patch Antenna for X-Band Applications”. In 3rd IEEE International Conference on Recent Trends in Electronics, Information & Communication Technology (RTEICT), pp. 411-413, May, 2018.‏

Finally, considering all of the foregoing points, the work lacks the requisite novelty.

Reviewer 4 Report

Efforts are appreciated but the design philosophy is not novel.